# Unsupervised Feature Learning with Emergent Data-Driven Prototypicality

## Abstract

Given a set of images, our goal is to map each image to a point in a feature space such that, not only point proximity indicates visual similarity, but where it is located directly encodes how prototypical the image is according to the dataset.

Our key insight is to perform unsupervised feature learning in hyperbolic instead of Euclidean space, where the distance between points still reflects image similarity, yet we gain additional capacity for representing prototypicality with the location of the point: The closer it is to the origin, the more prototypical it is. The latter property is simply emergent from optimizing the metric learning objective: The image similar to many training instances is best placed at the center of corresponding points in Euclidean space, but closer to the origin in hyperbolic space.

We propose an unsupervised feature learning algorithm in Hyperbolic space with sphere pACKing. HACK first generates uniformly packed particles in the Poincaré ball of hyperbolic space and then assigns each image uniquely to a particle. With our feature mapper simply trained to spread out training instances in hyperbolic space, we observe that images move closer to the origin with congealing - a warping process that aligns all the images and makes them appear more common and similar to each other, validating our idea of unsupervised prototypicality discovery. We demonstrate that our data-driven prototypicality provides an easy and superior unsupervised instance selection to reduce sample complexity, increase model generalization with atypical instances and robustness with typical ones.

## 1 Introduction

Not all instances are created equally. Some instances are more representative of the data set, whereas others are outliers or anomalies. Representative instances can be viewed as prototypes and used for interpretable machine learning (Bien & Tibshirani, 2011), curriculum learning (Bengio et al., 2009), and learning better decision boundaries (Carlini et al., 2018). Prototypes also allow us to classify with as few as or even one example (Miller et al., 2000). Given a set of images, thus it is desirable to organize them based on prototypicality to form a visual hierarchy.

If the image feature is given, it is relatively easy to find prototypes: We just need to identify density peaks of the feature distribution of the image set. Otherwise, discovering prototypical instances without supervision is difficult: There is no universal definition or simple metric to assess the prototypicality of the examples.

A naive method to address this problem is to examine the gradient magnitude (Carlini et al., 2018). However, this approach is shown to have a high variance which is resulted from different training setups (Carlini et al., 2018). Some methods address this problem from the perspective of adversarial robustness (Stock & Cisse, 2018; Carlini et al., 2018): prototypical examples should be more adversarially robust. However, the selection of the prototypical examples highly depends on the adversarial method and the metric used in the adversarial attack. Several other methods exist for this problem but they are either based on heuristics or lack a proper justification (Carlini et al., 2018).

Naturally, given a feature space, prototypical examples can be identified as density peaks. However, prototypicality undergoes changes as the feature space undergoes changes. In this paper, we propose an unsupervised feature learning algorithm, called HACK, for learning features that reflect prototypicality.In particular, HACK constructs a hierarchical arrangement of all the samples, with

typical examples positioned at the top level and atypical examples residing at the lower levels of the hierarchy. Different from existing unsupervised learning methods, HACK naturally leverages the geometry of *hyperbolic space* for unsupervised learning. Hyperbolic space is non-Euclidean space with constant non-negative curvature (Anderson, 2006). Different from Euclidean space, hyperbolic space can represent hierarchical relations with low distortion. Poincaré ball model is one of the most commonly used models for hyperbolic space (Nickel & Kiela, 2017b). One notable property of Poincaré ball model is that the distance to the origin grows exponentially as we move towards the boundary. Thus, the points located in the center of the ball are close to all the other points while the points located close to the boundary are infinitely far away from other points. With unsupervised learning in hyperbolic space, HACK can learn features which capture both visual similarity and hierarchical arrangements of the samples.

HACK optimizes the organization of the dataset by assigning the images to a set of uniformly distributed particles in hyperbolic space. The assignment is done by minimizing the total hyperbolic distance between the features and the particles via the Hungarian algorithm. The prototypicality arises naturally based on the distance of the example to the others. Prototypical examples tend to locate in the center of the Poincaré ball and atypical examples tend to locate close to the boundary. Hyperbolic space readily facilitates such an organization due to the property of the hyperbolic distance.

Our paper makes the following contributions.

- We propose the first unsupervised feature learning method to learn features which capture both visual similarity and prototypicality. The positions of the features reflect prototypicality of the examples.
- The proposed method HACK assigns images to particles that are uniformly packed in hyperbolic space. HACK fully exploits the property of hyperbolic space to construct a hierarchy of the samples in an unsupervised manner.
- We ground the concept of prototypicality based on congealing which conforms to human visual perception. The congealed examples can be used to replace the original examples for constructing datasets with known prototypicality. We validate the effectiveness of the method by using synthetic data with natural and congealed images. We further apply the proposed method to commonly used image datasets to reveal prototypicality.
- The discovered prototypical and atypical examples are shown to reduce sample complexity and increase the robustness of the model.

## 2 RELATED WORK

**Prototypicality.** The study of prototypical examples in machine learning has a long history. In Zhang (1992), the authors select typical instances based on the fact that typical instances should be representative of the cluster. In Kim et al. (2016), prototypical examples are defined as the examples that have maximum mean discrepancy within the data. Li et al. (Li et al., 2018) propose to discover prototypical examples by architectural modifications: project the dataset onto a low-dimensional manifold and use a prototype layer to minimize the distance between inputs and the prototypes on the manifold. The robustness to adversarial attacks is also used as a criterion for prototypicality (Stock & Cisse, 2018). In Carlini et al. (2018), the authors propose multiple metrics for prototypicality discovery. For example, the features of prototypical examples should be consistent across different training setups. However, these metrics usually depend heavily on the training setups and hyperparameters. The idea of prototypicality is also extensively studied in meta-learning for one-shot or few-shot classification (Snell et al., 2017). No existing works address the prototypicality discovery problem in a data-driven fashion. Our proposed HACK naturally exploits hyperbolic space to organize the images based on prototypicality.

**Unsupervised Learning in Hyperbolic Space.** Learning features in hyperbolic space have shown to be useful for many machine learning problems (Nickel & Kiela, 2017a; Ganea et al., 2018). One useful property is that hierarchical relations can be embedded in hyperbolic space with low distortion (Nickel & Kiela, 2017a). Wrapped normal distribution, which is a generalized version of the normal distribution for modeling the distribution of points in hyperbolic space (Nagano et al., 2019), is used as the latent space for constructing hyperbolic variational autoencoders (VAEs) (Kingma & Welling, 2013). Poincaré VAEs is constructed in Mathieu et al. (2019) with a similar idea to Nagano et al. (2019) by replacing the standard normal distribution with hyperbolic normal distribution.

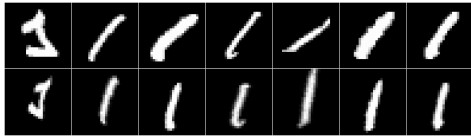

Figure 1: **Congealed images are more typical than the original images.** First row: sampled original images. Second row: the corresponding congealed images.

Unsupervised 3D segmentation (Hsu et al., 2020) and instance segmentation (Weng et al., 2021) are conducted in hyperbolic space via hierarchical hyperbolic triplet loss. CO-SNE (Guo et al., 2021a) is recently proposed to visualize high-dimensional hyperbolic features in a two-dimensional hyperbolic space. Although hyperbolic distance facilitates the learning of hierarchical structure, how to leverage hyperbolic space for unsupervised prototypicality discovery is not explored in the current literature.

## 3 SAMPLE HIERARCHY

**Sample Hierarchy VS. Class Hierarchy.** While most of the existing works in hierarchical image classification are focusing on using label hierarchy (Dhall et al., 2020; Guo et al., 2018), there also exists a natural hierarchy among different samples. In Khrulkov et al. (2020), the authors conducted an experiment to measure the $\delta$-hyperbolicity of the various image datasets and showed that common image datasets such as CIFAR10 and CUB exhibits natural hierarchical structure among the samples. Amongst a collection of images representing digit 1, suppose $\mathbf{x}$ is used for representing an image with a digit '1' that is upright, $\mathbf{x}'$ is used for representing an image with a digit 1 that leaning left and $\mathbf{x}''$ is used for representing an image with a digit '1' that leaning right. Given a metric $d(\cdot, \cdot)$, if we assume that $d(\mathbf{x}'', \mathbf{x}') \approx d(\mathbf{x}'', \mathbf{x}) + d(\mathbf{x}', \mathbf{x})$, in this context, we can naturally view the sample $\mathbf{x}$ as the root, and consider the other samples as its children in an underlying tree.

Compared with class hierarchy which can be extracted based on the pre-defined label relations, sample hierarchy is much harder to construct due to the lack of ground truth. Once a sample hierarchy is established, there are currently no existing methods available for verifying the accuracy of the hierarchy. Additionally, just like with class hierarchies, there may be ambiguities when constructing a sample hierarchy since multiple samples could potentially serve as the root.

**Building Sample Hierarchy from Density Peaks.** Given existing features $\{f(v_i)\}$ obtained by applying a feature extractor for each instance $v_i$, prototypical examples can be found by examining the density peaks via techniques from density estimation. For example, the K-nearest neighbor density (K-NN) estimation (Fix & Hodges, 1989) is defined as $p_{knn}(v_i, k) = \frac{k}{n} \frac{1}{A_d \cdot D^d(v_i, v_{k(i)})}$, where $d$ is the feature dimension, $A_d = \pi^{d/2}/\Gamma(d/2+1)$, $\Gamma(x)$ is the Gamma function and $k(i)$ is the $k$th nearest neighbor of example $v_i$. The nearest neighbors can be found by computing the distance between the features. Therefore, the process of constructing sample hierarchy through density estimation can be conceptualized as a two-step procedure involving: 1) feature learning and 2) detecting density peaks.

In the density estimation approach outlined above, the level of prototypicality depends on the learned features. Varying training setups can induce diverse feature spaces, resulting in differing conclusions on prototypicality. Nevertheless, prototypicality is an inherent attribute of the dataset and should remain consistent across various features. The aim of this paper is to extract features that intrinsically showcase the hierarchical organization of the samples. Specifically, by examining the feature alone within the feature space, we should be able to identify the example's prototypicality.

**Construct a Sample Hierarchy from Congealing.** To determine whether the feature truly captures prototypicality, it is necessary to identify which sample is the prototype. We ground our concept of prototypicality based on congealing (Miller et al., 2000). In particular, we define prototypical examples in the *pixel space* by examining the distance of the images to the average image in the corresponding class. Our idea is based on a traditional computer vision technique called image alignment (Szeliski et al., 2007) that aims to find correspondences across images. During congealing (Miller et al., 2000), a set of images are transformed to be jointly aligned by minimizing the joint pixel-wise entropies. The congealed images are more prototypical: they are better aligned with the

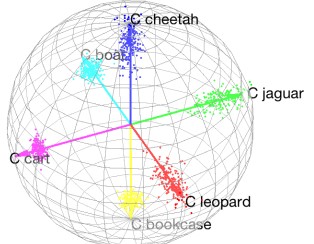 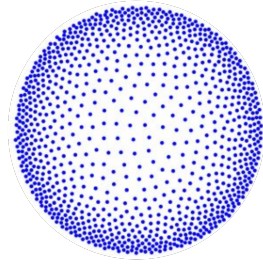 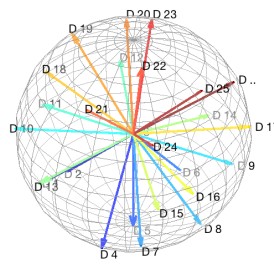

**a)** Supervised classification with fixed known targets

**b)** Our unsupervised feature learning with fixed but *unknown* targets

**c)** Metric feature learning with unknown targets

Figure 2: **The proposed HACK has a predefined geometrical arrangement and allows the images to be freely assigned to any particle.** a) Standard supervised learning has predefined targets. The image is only allowed to be assigned to the corresponding target. b) HACK packs particles uniformly in hyperbolic space to create initial seeds for the organization. The images are assigned to the particles based on their prototypicality and semantic similarities. c) Standard unsupervised learning has no predefined targets and images are clustered based on their semantic similarities.

average image. Thus, we have a simple way to transform an atypical example into a typical example (see Figure 1). This is useful since given an unlabeled image dataset the typicality of the examples is unknown, congealing examples can be naturally served as examples with known typicality and be used as a validation for the effectiveness of our method.

## 4 UNSUPERVISED HYPERBOLIC FEATURE LEARNING

### 4.1 POINCARÉ BALL MODEL FOR HYPERBOLIC SPACE

**Poincaré Ball Model for Hyperbolic Space.** Euclidean space has a curvature of zero and a hyperbolic space is a Riemannian manifold with constant negative curvature.. There are several isometrically equivalent models for visualizing hyperbolic space with Euclidean representation. The Poincaré ball model is the commonly used one in hyperbolic representation learning (Nickel & Kiela, 2017b). The $n$-dimensional Poincaré ball model is defined as $(\mathbb{B}^n, \mathfrak{g}_{\mathbf{x}})$, where $\mathbb{B}^n = \{\mathbf{x} \in \mathbb{R}^n : \|\mathbf{x}\| < 1\}$ and $\mathfrak{g}_{\mathbf{x}} = (\gamma_{\mathbf{x}})^2 I_n$ is the Riemannian metric tensor. $\gamma_{\mathbf{x}} = \frac{2}{1-\|\mathbf{x}\|^2}$ is the conformal factor and $I_n$ is the Euclidean metric tensor.

**Hyperbolic Distance.** Given two points $\boldsymbol{u} \in \mathbb{B}^n$ and $\boldsymbol{v} \in \mathbb{B}^n$, the hyperbolic distance is defined as,

$$d_{\mathbb{B}^n}(\boldsymbol{u}, \boldsymbol{v}) = \text{arcosh}\left(1 + 2\frac{\|\boldsymbol{u} - \boldsymbol{v}\|^2}{(1 - \|\boldsymbol{u}\|^2)(1 - \|\boldsymbol{v}\|^2)}\right) \tag{1}$$

where $\text{arcosh}$ is the inverse hyperbolic cosine function and $\|\cdot\|$ is the usual Euclidean norm.

Hyperbolic distance has the unique property that it grows exponentially as we move towards the boundary of the Poincaré ball. In particular, the points on the circle represent points in infinity. Hyperbolic space is naturally suitable for embedding hierarchical structure (Sarkar, 2011; Nickel & Kiela, 2017b) and can be regarded as a continuous representation of trees (Chami et al., 2020). The hyperbolic distance between samples implicitly reflects their hierarchical relation. Thus, by embedding images in hyperbolic space we can naturally organize images based on their semantic similarity and prototypicality.

### 4.2 BUILDING SAMPLE HIERARCHY IN HYPERBOLIC SPACE

Hyperbolic space is naturally suitable for embedding tree structure. However, in order to leverage hyperbolic space to build a sample hierarchy in an unsupervised manner, a suitable objective function is still missing. There are two challenges in designing the objective function. First, the underlying tree structure of the samples is unknown. Second, how to perform feature learning such that hierarchy can naturally emerge is unclear.

To address the first challenge, instead of creating a predefined tree structure that might not faithfully represent the genuine hierarchical organization, we leverage sphere packing in hyperbolic space for building a skeleton for placing the samples. We specify where the particles should be located ahead of training with uniform packing, which by design are maximally evenly spread out in hyperbolic space. The uniformly distributed particles guide feature learning to achieve maximum instance discrimination (Wu et al., 2018) while enabling us to extract a tree structure from the samples.

To address the second challenge, HACK figures out which instance should be mapped to which target through bipartite graph matching as a global optimization procedure. During training, HACK minimizes the total hyperbolic distances between the mapped image point (in the feature space) and the target, those that are more typical naturally emerge closer to the origin of Poincaré ball. HACK differs from the existing learning methods in several aspects (Figure 2). Different from supervised learning, HACK allows the image to be assigned to *any* target (particle). This enables the exploration of the natural organization of the data. Different from unsupervised learning method, HACK specifies a predefined geometrical organization which encourages the corresponding structure to be emerged from the dataset.

## 4.3 SPHERE PACKING IN HYPERBOLIC SPACE

Given $n$ particles, our goal is to pack the particles into a two-dimensional hyperbolic space as densely as possible. We derive a simple repulsion loss function to encourage the particles to be equally distant from each other. The loss is derived via the following steps. First, we need to determine the radius of the Poincaré ball used for packing. We use a curvature of 1.0 so the radius of the Poincaré ball is 1.0. The whole Poincaré ball cannot be used for packing since the volume is infinite. We use $r < 1$ to denote the actual radius used for packing. Thus, our goal is to pack $n$ particles in a compact subspace of Poincaré ball. Then, the Euclidean radius $r$ is further converted into hyperbolic radius $r_{\mathbb{B}}$. Let $s = \frac{1}{\sqrt{c}}$, where $c$ is the curvature. The relation between $r$ and $r_{\mathbb{B}}$ is $r_{\mathbb{B}} = s \log \frac{s+r}{s-r}$. Next, the total hyperbolic area $A_{\mathbb{B}}$ of a Poincaré ball of radius $r_{\mathbb{B}}$ can be computed as $A_{\mathbb{B}} = 4\pi s^2 \sinh^2(\frac{r_{\mathbb{B}}}{2s})$, where $\sinh$ is the hyperbolic sine function. Finally, the area per point $A_n$ can be easily computed as $\frac{A_{\mathbb{B}}}{n}$, where $n$ is the total number of particles. Given $A_n$, the radius per point can be computed as $r_n = 2s \sinh^{-1}(\sqrt{\frac{A_n}{4\pi s^2}})$. We use the following loss to generate uniform packing in hyperbolic space. Given two particles $i$ and $j$, the repulsion loss $V$ is defined as,

$$V(i, j) = \{\frac{1}{[2r_n - \max(0, 2r_n - d_{\mathbb{B}}(i, j))]^k} - \frac{1}{(2r_n)^k}\} \cdot C(k) \tag{2}$$

where $C(k) = \frac{(2r_n)^{k+1}}{k}$ and $k$ is a hyperparameter. Intuitively, if the particle $i$ and the particle $j$ are within $2r_n$, the repulsion loss is positive. Minimizing the repulsion loss would push the particles $i$ and $j$ away. If the repulsion is zero, this indicates all the particles are equally distant. Also the repulsion loss grows significantly when two particles become close.

We also adopt the following boundary loss to prevent the particles from escaping the ball,

$$B(i; r) = \max(0, \text{norm}_i - r + \text{margin}) \tag{3}$$

where $\text{norm}_i$ is the $\ell_2$ norm of the representation of the particle $i$. Figure 2 b) shows an example of the generated particles that are uniformly packed in hyperbolic space.

## 4.4 HYPERBOLIC INSTANCE ASSIGNMENT

HACK learns the features by optimizing the assignments of the images to particles (Figure 3). The assignment should be one-to-one, i.e., each image should be assigned to one particle and each particle is allowed to be associated with one image. We cast the instance assignment problem as a bipartite matching problem (Gibbons, 1985) and solve it with Hungarian algorithm (Munkres, 1957).

Initially, we randomly assign the particles to the images, thus there is a random one-to-one correspondence between the images to the particles (not optimized). Given a batch of samples $\{(\mathbf{x}_1, \mathbf{s}_1), (\mathbf{x}_2, \mathbf{s}_2), ..., (\mathbf{x}_B, \mathbf{s}_B)\}$, where $\mathbf{x}_i$ is an image and $\mathbf{s}_i$ is the corresponding particle, and an encoder $f_\theta$, we generate the hyperbolic feature for each image $\mathbf{x}_i$ as $f_\theta(\mathbf{x}_i) \in \mathbb{B}^2$, where $\mathbb{B}^2$ is a two-dimensional Poincaré ball. For a given hyperbolic feature $f_\theta(\mathbf{x})$, with fixed particle locations,

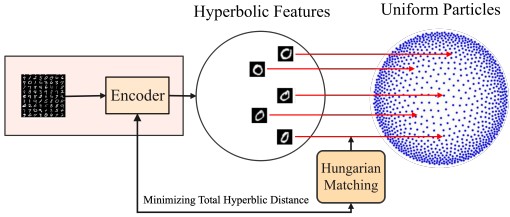

Figure 3: **HACK conducts unsupervised learning in hyperbolic space with sphere packing.** The images are mapped to particles by minimizing the total hyperbolic distance. HACK learns features that can capture both visual similarities and prototypicality.

**Algorithm 1** HACK: Unsupervised Learning in Hyperbolic Space.

---

**Require:** # of images: $n \geq 0$. Radius for packing: $r < 1$. An encoder with parameters $\theta$: $f_\theta$
1: Generate uniformly distributed particles in hyperbolic space by minimizing the repulsion loss in Equation 2
2: Given $\{(\mathbf{x}_1, s_1), (\mathbf{x}_2, s_2), ..., (\mathbf{x}_b, s_b)\}$, optimize $f_\theta$ by minimizing the total hyperbolic distance via Hungarian algorithm.

---

the distance between the hyperbolic feature and the particles signifies the hierarchical level of the associated sample. Thus, to determine the hierarchical levels for all samples within the hierarchy, we must establish a one-to-one mapping between all the samples and the particles. This can be cast as the following bipartite matching problem in hyperbolic space,

$$\ell(\theta, \pi) = \sum_{i=1}^{B} d_{\mathbb{B}^n}\big(f_\theta(\mathbf{x}_i), \mathbf{s}_{\pi(f_\theta(\mathbf{x}_i))}\big) \tag{4}$$

where $\pi : f_\theta(\mathbf{x}) \to \mathbb{N}$ is a projection function which projects hyperbolic features to a particle index. Minimizing $\ell(\theta, \pi)$ with respect to $\pi$ is a combinatorial optimization problem, which can hardly be optimized with $\theta$ using gradient-based algorithms. Thus, we adopt a joint optimization strategy which optimizes $\theta$ and $\pi$ alternatively. In each batch, we first leverage the Hungarian algorithm (Munkres, 1957) to find the optimal matching $\pi^*$ based on the current hyperbolic features. Then we minimize Eq. 4 with respect to $\theta$ based on the current assignment $\pi^*$. This process is repeated for a certain number of epochs until convergence is achieved.

The Hungarian algorithm (Munkres, 1957) has a complexity of $\mathcal{O}(x^3)$, where $x$ is the number of items. As we perform the particle assignment in the batch level, the time and memory complexity is tolerable. Also, the one-to-one correspondence between the images and particles is always maintained during training. After training, based on the assigned particle, the level of the sample in the hierarchy can be easily retrieved. The details of HACK are shown in Algorithm 1.

## 5 EXPERIMENTS

We design several experiments to show the effectiveness of HACK for the semantic and hierarchical organization. First, we first construct a dataset with known hierarchical structure using the congealing algorithm (Miller et al., 2000). Then, we apply HACK to datasets with unknown hierarchical structure to organize the samples based on the semantic and prototypical structure. Finally, we show that the prototypical structure can be used to reduce sample complexity and increase model robustness. **Datasets.** We first construct a dataset called *Congealed MNIST*. To verify the efficacy of HACK for unsupervised prototypicality discovery, we need a benchmark with known prototypical examples. However, currently there is no standard benchmark for this purpose. To construct the benchmark, we use the congealing algorithm from Miller et al. (2000) to align the images in each class of MNIST (LeCun, 1998). The congealing algorithm is initially used for one-shot classification. During congealing, the images are brought into correspondence with each other jointly. The congealed images are more prototypical: they are better aligned with the average image. The synthetic data is generated by replacing 500 original images with the corresponding congealed images. In Section E of the Appendix, we show the results of changing the number of replaced original images. We expect HACK to discover the congealed images and place them in the center of the Poincaré ball. We also aim to discover the prototypical examples from each class of the standard MNIST dataset (LeCun, 1998) and CIFAR10 (Krizhevsky et al., 2009). CIFAR10 consists of 60000 from 10 object categories ranging from airplane to truck. CIFAR10 is more challenging than MNIST since it has larger intra-class variations.

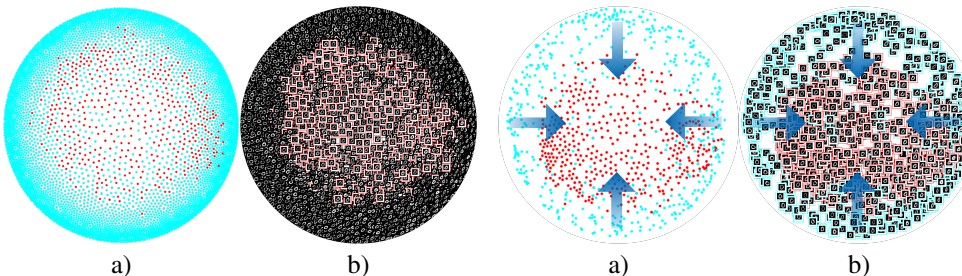

a)          b)          a)          b)

Figure 5: **Congealed images are located in the center of the Poincaré ball.** a) Red dots denote congealed images and cyan dots denote original images. b) Typical images are in the center and atypical images are close to the boundary. Images are also clustered together based on visual similarity. Congealed images are shown in red boxes.

Figure 6: **Original images are pushed to the center of the ball after congealing.** We train the first model with original images. Then we train the second model by replacing a subset of original images (marked with cyan) with the corresponding congealed images. The features of the congealed images (marked with red) become closer to the center of the ball.

**Baselines.** We consider several existing metrics proposed in Carlini et al. (2018) for prototypicality discovery, the details can be found in Section C of the Appendix.

- Holdout Retraining (Carlini et al., 2018): We consider the Holdout Retraining proposed in Carlini et al. (2018). The idea is that the distance of features of prototypical examples obtained from models trained on different datasets should be close.
- Model Confidence (Carlini et al., 2018): Intuitively, the model should be confident in prototypical examples. Thus, it is natural to use the confidence of the model prediction as the criterion for prototypicality.
- UHML (Yan et al., 2021): UHML is an unsupervised hyperbolic learning method which aims to discover the natural hierarchies of data by taking advantage of hyperbolic metric learning and hierarchical clustering.

**Implementation Details.** We implement HACK in PyTorch and the code will be made public. To generate uniform particles, we first randomly initialize the particles and then run the training for 1000 epochs with a 0.01 learning rate to minimize the repulsion loss and boundary loss. The curvature of the Poincaré ball is 1.0 and the $r$ is 0.76 which is used to alleviate the numerical issues (Guo et al., 2021b). The hyperparameter $k$ is 1.55 which is shown to generate uniform particles well. For the assignment, we use a LeNet (LeCun et al., 1998) for MNIST and a ResNet20 (He et al., 2016) for CIFAR10 as the encoder. We apply HACK to each class separately. We attach a fully connected layer to project the feature into a two-dimensional Euclidean space. The image features are then projected onto hyperbolic space via an exponential map. We run the training for 200 epochs using a cosine learning rate scheduler (Loshchilov & Hutter, 2016) with an initial learning rate of 0.1. We optimize the assignment *every other* epoch. All the experiments are run on an NVIDIA TITAN RTX GPU.

## 5.1 PROTOTYPICALITY IN THE HYPERBOLIC FEATURE NORM

We explicitly show that the hyperbolic space can capture prototypicality by analyzing the relation between hyperbolic norms and the K-NN density estimation. Taking the learned hyperbolic features, we first divide the range of norms of hyperbolic features into numerous portions with equal length (50 portions for this plot). The mean K-NN density is calculated by averaging the density estimation of features within each portion. Figure 4 shows that the mean density drops as the norm increases, which shows that the prototypicality emerges automatically within the norms, the inherent characteristic of hyperbolic space. This validates that prototypicality is reflected in the hyperbolic feature norm.

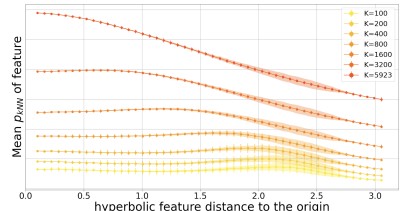

Figure 4: **Hyperbolic space can capture the prototypicality inherently.** The error bar of each point is given by the variance of density within the corresponding portion, and the width of the shaded band indicates the number of features within the portion.

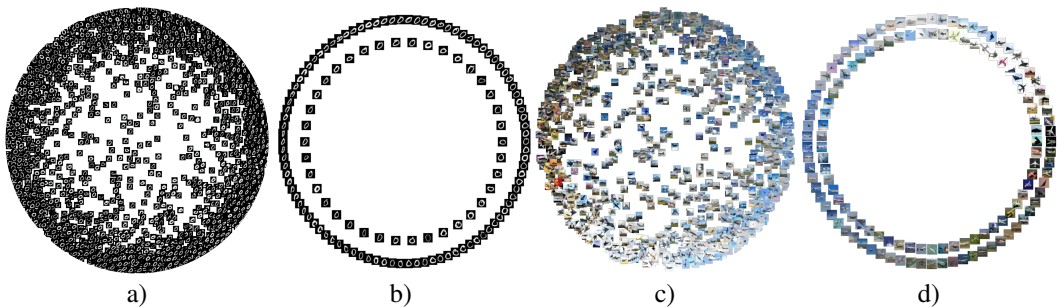

a)  b)  c)  d)

Figure 8: **Our unsupervised learning methods conform to our visual perception**. a) Samples of 2000 images from MNIST. b) Images of MNIST arranged angularly. c) Samples of 2000 images from CIFAR10. d) Images of CIFAR10 arranged angularly. Images are organized based on prototypicality and visual similarity.

## 5.2 Visual Prototypicality: Congealed MNIST

We further apply HACK for visual feature learning on congealed MNIST. Figure 5 shows that HACK can discover the congealed images from all images. In Figure 5 a), the red particles denote the congealed images and cyan particles denote the original images. We can observe that the congealed images are assigned to the particles located in the center of the Poincaré ball. This verifies that HACK can *indeed* discover prototypical examples from the original dataset. Section G.1 in the Appendix shows that the features of atypical examples gradually move to the boundary of the Poincaré ball during training. In Figure 5 b), we show the actual images that are embedded in the two-dimensional hyperbolic space. We can observe that the images in the center of Poincaré ball are more prototypical and images close to the boundary are more atypical. Also, the images are naturally organized by their semantic similarity. Figure 6 shows that the features of the original images become closer to the center of Poincaré ball after congealing. In summary, HACK can discover prototypicality and also organize the images based on their semantic and hierarchical structure. To the best of our knowledge, this is the first unsupervised learning method that can be used to discover prototypical examples in a data-driven fashion.

## 5.3 Prototypicality for Instance Selection

Figure 8 shows the embedding of class 0 from MNIST and class "airplane" from CIFAR10 in the hyperbolic space. We sample 2000 images from MNIST and CIFAR10 for better visualization. We also show the arrangement of the images angularly with different angles. Radially, we can observe that images are arranged based on prototypicality. The prototypical images tend to be located in the center of the Poincaré ball. Especially for CIFAR10, the images become blurry and even unrecognizable as we move toward the boundary of the ball. Angularly, the images are arranged based on visual similarity. The visual similarity of images has a smooth transition as we move around angularly. Please see Section D in the Appendix for more results.

**Comparison with Baselines.** Figure 7 shows the comparison of the baselines with HACK . We can observe that both HACK and Model Confidence (MC) can discover typical and atypical images. Compared with MC, HACK defines prototypicality as the distance of the sample to other samples which is more aligned with human intuition. Moreover, in addition to prototypicality, HACK can also be used to organize examples by semantic similarities. Holdout Retraining (HR) is not effective for prototypicality discovery due to the randomness of model training.

Figure 7: **HACK can better identify typical and atypical examples compared with HR and MR.**

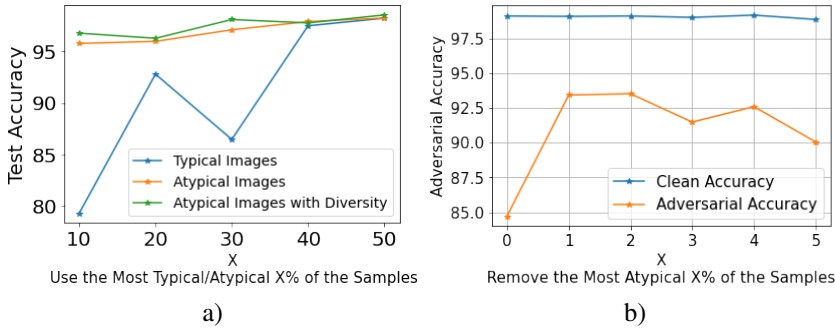

Figure 9: **HACK can be used to construct sample hierarchy which is useful for several downstream tasks.** a) Training with atypical examples achieves higher accuracy than training with typical examples. b) The adversarial accuracy greatly improves after removing the X% of most atypical examples.

## 5.4 APPLICATION OF PROTOTYPICALITY

**Reducing Sample Complexity.** The proposed HACK can discover prototypical images as well as atypical images. We show that with *atypical* images we can reduce the sample complexity for training the model. Prototypical images are representative of the dataset but lack variations. Atypical examples contain more variations and it is intuitive that models trained on atypical examples should generalize better to the test samples. To verify this hypothesis, we select a subset of samples based on the norm of the features which indicates prototypicality of the examples. In particular, typical samples correspond to the samples with smaller norms and atypical samples correspond to the samples with larger norms. The angular layout of the hyperbolic features naturally captures sample diversity, thus for selecting atypical examples, we also consider introducing more diversity by sampling images with large norms along the angular direction.

We train a LeNet on MNIST for 10 epochs with a learning rate of 0.1. Figure 9 a) shows that training with atypical images can achieve much higher accuracy than training with typical images. In particular, training with the most atypical 10% of the images achieves 16.54% higher accuracy than with the most typical 10% of the images. Thus, HACK provides an easy solution to reduce sample complexity. We also compared UHML (Yan et al., 2021), which is an unsupervised metric learning in hyperbolic space, with HACK on the MNIST dataset. By incorporating 10% atypical samples based on feature norm, HACK outperformed UHML by 10.2%. Also by excluding the 1% atypical examples, HACK achieved an additional 5.7% improvement over UHML.

**Increasing Model Robustness.** Training models with atypical examples can lead to a vulnerable model to adversarial attacks (Liu et al., 2018; Carlini et al., 2018). Intuitively, atypical examples lead to a less smooth decision boundary thus a small perturbation to examples is likely to change the prediction. With HACK, we can easily identify atypical samples to improve the robustness of the model. We use MNIST as the benchmark and use FGSM (Goodfellow et al., 2014) to attack the model with an $\epsilon = 0.07$. We identify the atypical examples with HACK and remove the most atypical X% of the examples. Figure 9 b) shows that discarding atypically examples greatly improves the robustness of the model: the adversarial accuracy is improved from 84.72% to 93.42% by discarding the most atypical 1% of the examples. It is worth noting that the clean accuracy remains the same after removing a small number of atypical examples.

## 6 SUMMARY

We propose an unsupervised learning method, called HACK, for organizing images with sphere packing in hyperbolic space. HACK optimizes the assignments of the images to a fixed set of uniformly distributed particles by naturally exploring the properties of hyperbolic space. As a result, prototypical and semantic structures emerge naturally due to feature learning. We apply HACK to synthetic data with known prototypicality and standard image datasets. The discovered prototypicality and atypical examples can be used to reduce sample complexity and increase model robustness. The idea of HACK can also be generalized to learn other geometrical structures from the data by specifying different geometric patterns.

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

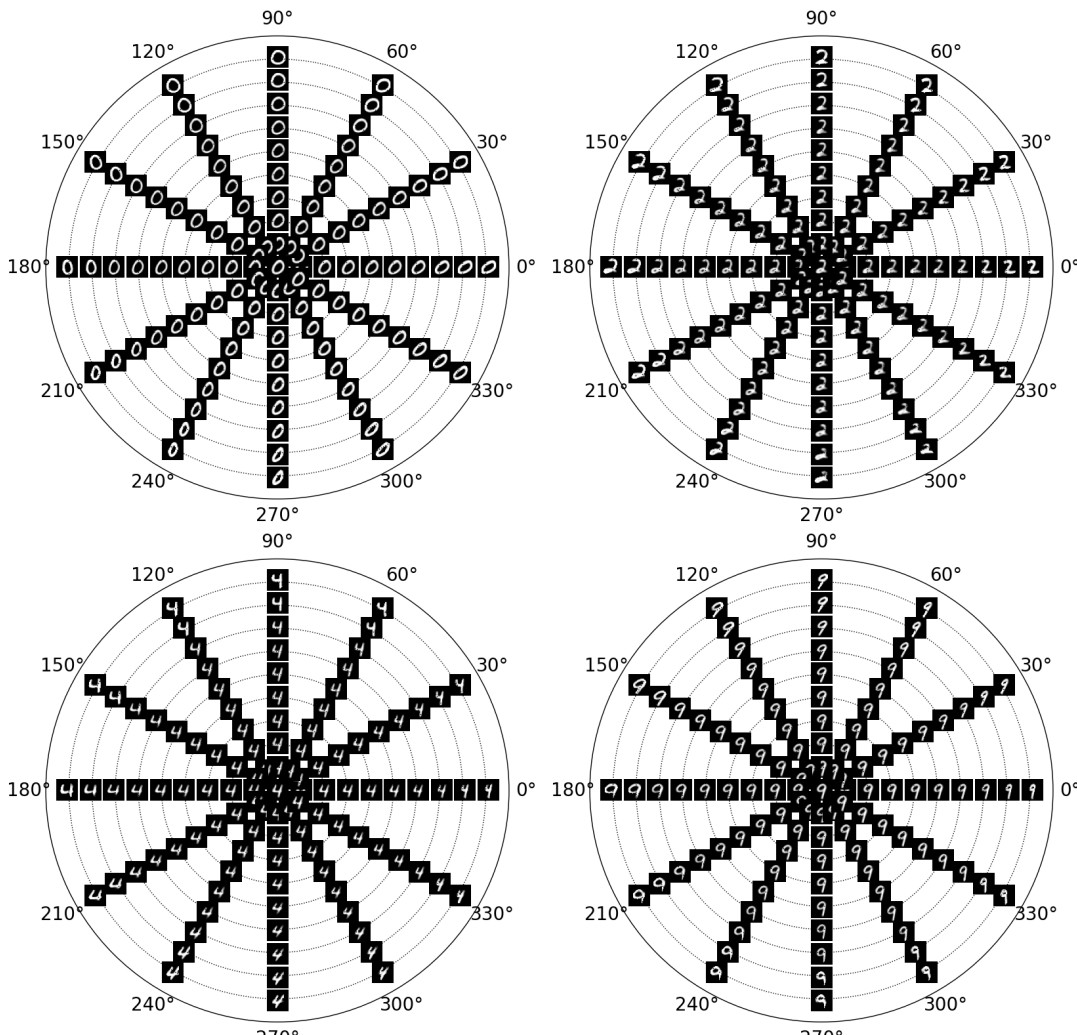

Figure 10: **Decoded images in feature space learned by HACK.** The decoder for every class is trained individually. The angles indicate the uniform division of the space. For each orientation, ten equidistant points were selected and fed into the decoder to generate corresponding images.

## A  HYPERBOLIC SPACE AS A CONTINUOUS TREE

As the Poincaré ball model can be regarded as a continuous representation of trees (Chami et al., 2020), every point in the space should correspond to a certain image. To verify the feature space learned by HACK can reflect the prototypicality, we train a simple MLP decoder with two hidden layers with the representations learned by HACK. Specifically, this decoder takes as input the 2-dimensional features in the Poincaré ball model and outputs an image. Using this decoder, we can get an image corresponding to every point in the Poincaré ball model.

Figure 10 shows the results of the decoder on a subset of MNIST digits. Initially, individual decoders were trained for each class. The entire space was then uniformly divided into 12 orientations. For each orientation, ten equidistant points were selected and fed into the decoder to generate corresponding images. It is evident that the images produced by the decoder exhibit a transition from typical to atypical representations across every orientation. Moreover, similar trends are observed for closely aligned orientations. For example, in the results for class 2, the imagery transitions from a typical representation at the origin to 12 distinct atypical forms. Notably, in the orientations spanning 120° to 270°, six showed a small loop at the base of the 2. Conversely, the remaining six orientations displayed a 2 with a straight base. The results suggest that the feature space learned by HACK

distinctly positions typical image characteristics closer to the center, while atypical image features are pushed to the boundary.

## B   MORE DETAILS ON HYPERBOLIC INSTANCE ASSIGNMENT

A more detailed description of the hyperbolic instance assignment is given.

Initially, we randomly assign the particles to the images. Given a batch of samples $\{(\mathbf{x}_1, s_1), (\mathbf{x}_2, s_2), ..., (\mathbf{x}_b, s_b)\}$, where $\mathbf{x}_i$ is an image and $s_i$ is the corresponding particle. Given an encoder $f_\theta$, we generate the hyperbolic feature for each image $\mathbf{x}_i$ as $f_\theta(\mathbf{x}_i) \in \mathbb{B}^2$, where $\mathbb{B}^2$ is a two-dimensional Poincaré ball.

we aim to find the minimum cost bipartite matching of the images to the particles. The cost to minimize is the total hyperbolic distance of the hyperbolic features to the particles. We first compute all the pairwise distances between the hyperbolic features and the particles. This is the cost matrix of the bipartite graph. Then we use the Hungarian algorithm to optimize the assignment (Figure 11).

Suppose we train the encoder $f_\theta$ for $T$ epochs. We run the hyperbolic instance assignment every other epoch to avoid instability during training. **We optimize the encoder $f_\theta$ to minimize the hyperbolic distance of the hyperbolic feature to the assigned particle in each batch**.

## C   DETAILS OF BASELINES

**Holdout Retraining:** We consider the Holdout Retraining proposed in Carlini et al. (2018). The idea is that the distance of features of prototypical examples obtained from models trained on different datasets should be close. In Holdout Retraining, multiple models are trained on the same dataset. The distances of the features of the images obtained from different models are computed and ranked. The prototypical examples are those examples with the closest feature distance.

**Model Confidence:** Intuitively, the model should be confident on prototypical examples. Thus, it is natural to use the confidence of the model prediction as the criterion for prototypicality. Once we train a model on the dataset, we use the confidence of the model to rank the examples. The prototypical examples are those examples that the model is most

## D   MORE RESULTS ON PROTOTYPICALITY DISCOVERY

We show the visualization of all the images in Figure 16 and Figure 17. The images are organized naturally based on their prototypicality and semantic similarity. We further conduct retrieval based on the norm of the hyperbolic features to extract the most typical and atypical images on CIAFR10 in Figure 18. The hyperbolic features with large norms correspond to atypical images and the hyperbolic features with small norms correspond to typical images. It can be observed that the object in the atypical images is not visible.

## E   GRADUALLY ADDING MORE CONGEALED IMAGES

We gradually increase the number of original images replaced by congealed images from 100 to 500. Still, as shown in Figure 12, HACK can learn a representation that captures the concept of prototypicality regardless of the number of congealed images. This again confirms the effectiveness of HACK for discovering prototypicality.

## F   DIFFERENT RANDOM SEEDS

We further run the assignment 5 times with different random seeds. The results are shown in Figure 13. We observe that the algorithm does not suffer from high variance and the congealed images are always assigned to the particles in the center of the Poincaré ball. This further confirms the efficacy of the proposed method for discovering prototypicality.

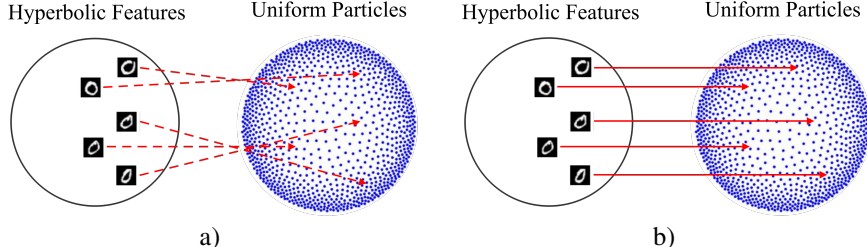

Figure 11: **Hyperbolic Instance Assignment minimizes the total hyperbolic distances between the image features and the particles.** a) Initial assignment. b) Optimized assignment.

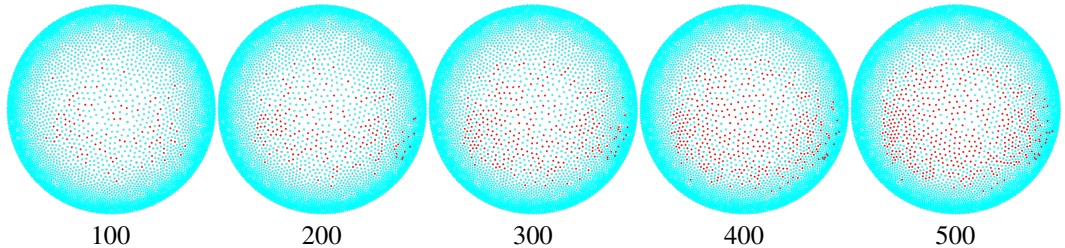

Figure 12: **HACK consistently places congealed images in the center of the Poincaré ball**. We gradually increase the number of original images replaced by congealed images from 100 to 500. The congealed images are marked with red dots and the original images are marked with cyan dots.

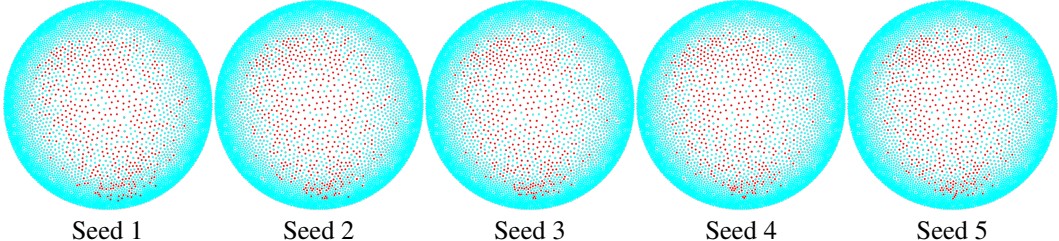

Figure 13: **HACK consistently places congealed images in the center of the Poincaré ball in multiple runs with different random seeds.**. The congealed images are marked with red dots and the original images are marked with cyan dots.

## G  EMERGENCE OF PROTOTYPICALITY IN THE FEATURE SPACE

Existing unsupervised learning methods mainly focus on learning features for differentiating different classes or samples Wu et al. (2018); He et al. (2020); Chen et al. (2020). The learned representations are transferred to various downstream tasks such as segmentation and detection. In contrast, the features learned by HACK aim at capturing prototypicality within a single class.

To investigate the effectiveness of HACK in revealing prototypicality, we can include or exclude congealed images in the training process. When the congealed images are included in the training process, we expect the congealed images to be located in the center of the Poincaré ball while the original images to be located near the boundary of the Poincaré ball. When the congealed images are excluded from the training process, we expect the features of congealed images produced via the trained network to be located in the center of the Poincaré ball.

### G.1  TRAINING WITH CONGEALED IMAGES AND ORIGINAL IMAGES

We follow the same setups as in Section 4.3.1 of the main text. Figure 14 shows the hyperbolic features of the congealed images and original images in different training epochs. The features of the congealed images stay in the center of the Poincaré ball while the features of the original images gradually expand to the boundary.

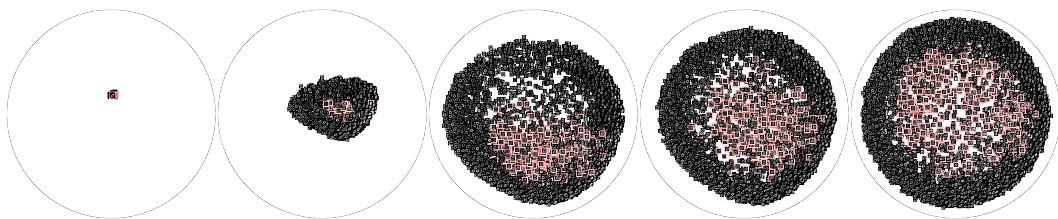

Epoch 1     Epoch 5     Epoch 10     Epoch 15     Epoch 200

Figure 14: **Atypical images gradually move to the boundary of the Poincaré ball**. This shows that the representations learned by HACK capture prototypicality. Congealed images are in red boxes which are more typical. The network is trained with *both* the congealed images and original images.

### G.2 TRAINING ONLY WITH ORIGINAL IMAGES

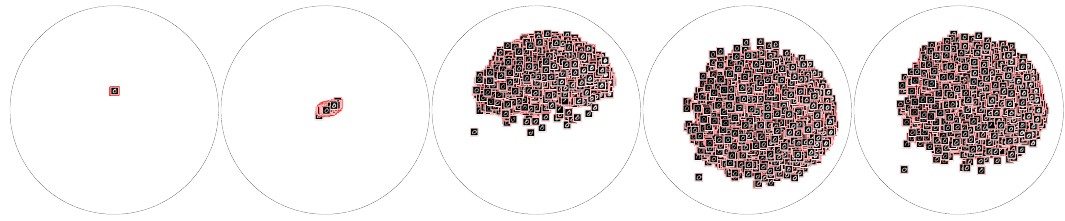

Epoch 1     Epoch 10     Epoch 20     Epoch 40     Epoch 200

Figure 15: **The representations learned by HACK gradually capture prototypicality during the training process.** Congealed images are in red boxes which are more typical. We produce the features of the congealed images with the trained network in different epochs. The network is *only* trained with original images.

Figure 15 shows the hyperbolic features of the congealed images **when the model is trained only with original images**. As we have shown before, congealed images are naturally more typical than their corresponding original images since they are aligned with the average image. The features of congealed images are all located close to the center of the Poincaré ball. This demonstrates that prototypicality naturally emerges in the feature space.

Without using congealed images during training, we exclude any artifacts and further confirm the effectiveness of HACK for discovering prototypicality. We also observe that the features produced by HACK also capture the fine-grained similarities among the congealing images despite the fact that all the images are aligned with the average image.

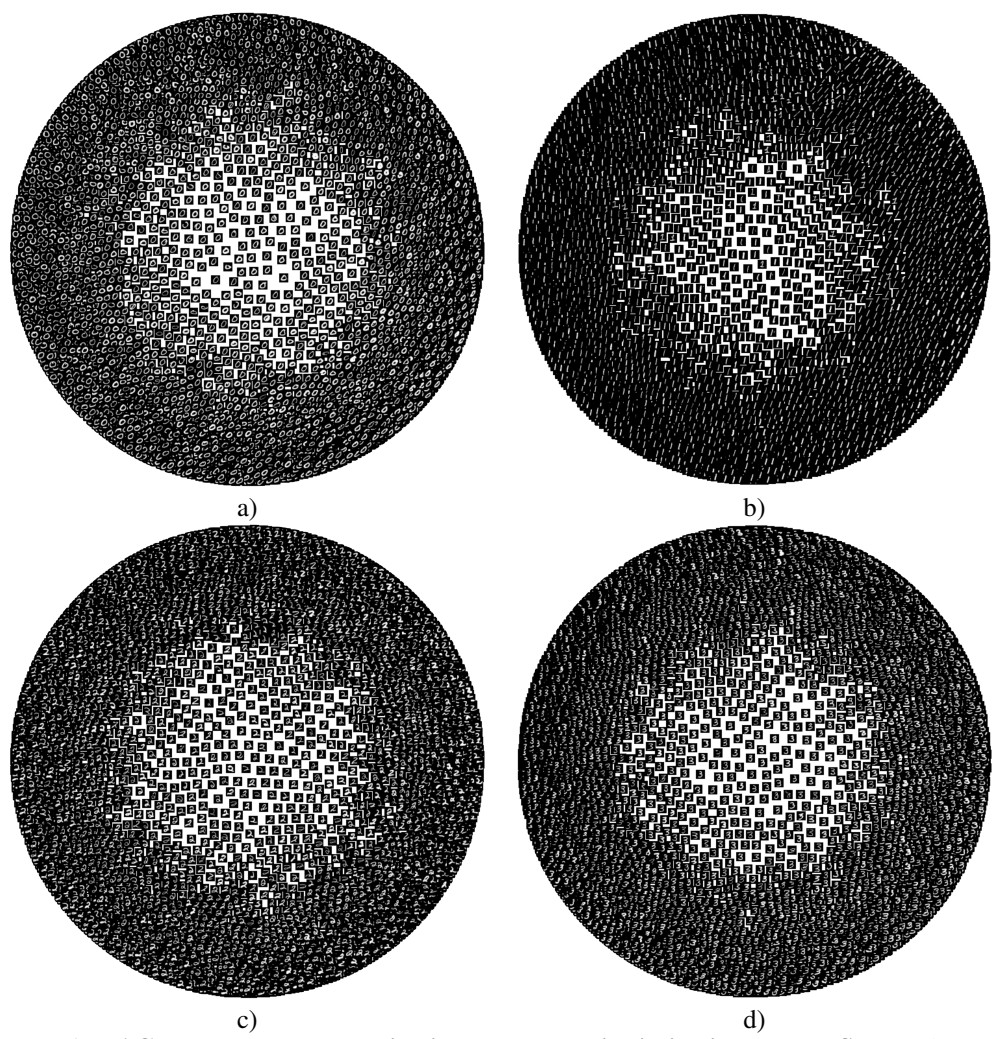

Figure 16: **HACK captures prototypicality and semantic similarity on MNIST.** a) Class 0. b) Class 1. c) Class 2. d) Class 3.

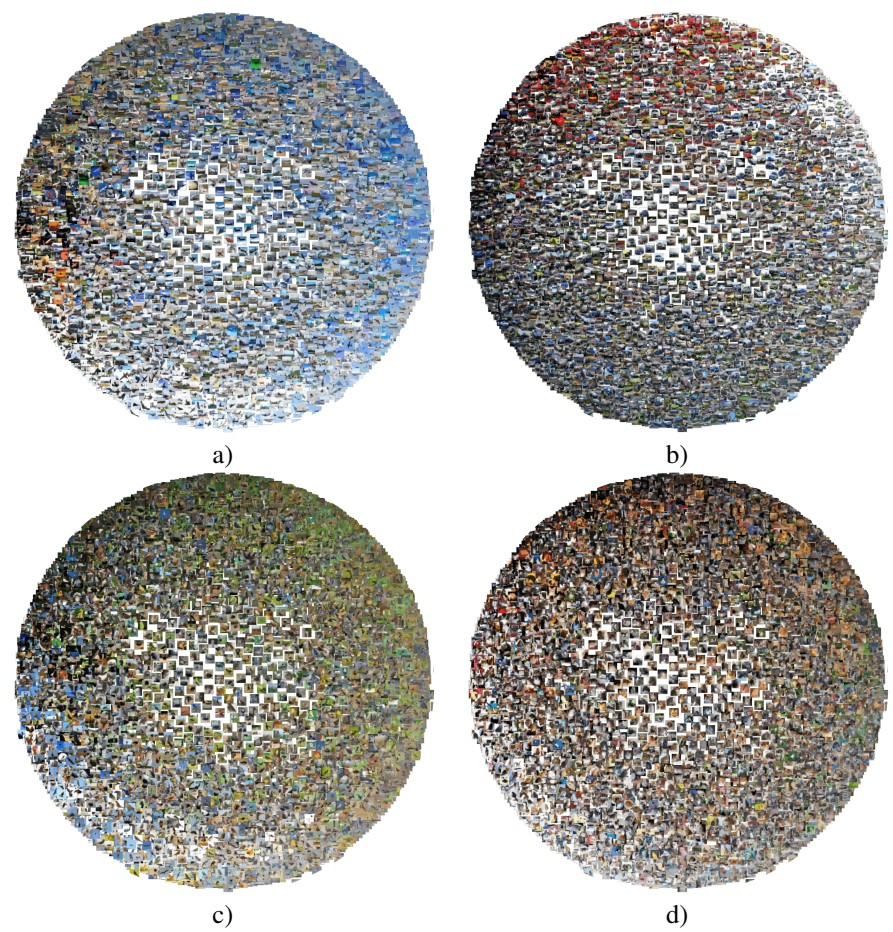

Figure 17: **HACK captures prototypicality and semantic similarity on CIFAR10.** a) Class "airplane". b) Class "automobile". c) Class "bird". d) Class "cat".

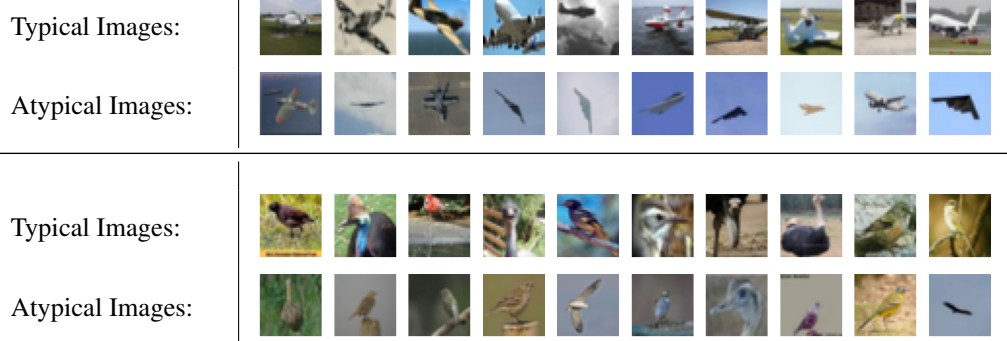

Figure 18: **Most typical and atypical images extracted by HACK from CIFAR10.**

## H  DISCUSSIONS ON SOCIETAL IMPACT AND LIMITATIONS.

We address the problem of unsupervised learning in hyperbolic space. We believe the proposed HACK should not raise any ethical considerations. We discuss current limitations below,

**Applying to the Whole Dataset** Currently, HACK is applied to each class separately. Thus, it would be interesting to apply HACK to all the classes at once without supervision. This is much more challenging since we need to differentiate between examples from different classes as well as the prototypical and semantic structure.

**Exploring other Geometrical Structures** We consider uniform packing in hyperbolic space to organize the images. It is also possible to extend HACK by specifying other geometrical structures to encourage the corresponding organization to emerge from the dataset.

