# OpenReview forum: "Unsupervised Feature Learning with Emergent Data-Driven Prototypicality"
_ICLR.cc/2024/Conference — ICLR 2024 Conference Withdrawn Submission_

### Official Review · Reviewer_9yFS · 2023-10-26

**Soundness:** 2 fair
**Presentation:** 2 fair
**Contribution:** 3 good
**Rating:** 3
**Confidence:** 4

**Summary:**

This paper proposes HACK for unsupervised learning that can arrange images in hyperbolic space. HACK optimizes image assignments to a fixed set of uniformly distributed particles in the hyperbolic space. It's found that the prototypicality property is emergent from such optimization: images similar to many training instances (more prototypical) are closer to the origin in hyperbolic space. The authors validate the effectiveness of HACK using synthetic data with natural and congealed images. They also test the method on the real MNIST and CIFAR datasets to reveal prototypicality. Lastly, the discovered prototypical and atypical examples are shown to reduce sample complexity and increase model robustness to some extent.

**Strengths:**

- The proposed unsupervised method HACK does have clear distinctions with existing methods: unlike supervised learning, HACK allows the image to be assigned to any target (particle). Unlike existing unsupervised learning method, HACK learns to match to
a predefined geometrical organization in hyperbolic space (uniformly distributed).
- The core instance assignment problem is cast as a bipartite matching problem and solved with the well-known Hungarian algorithm that has good convergence properties.
- Besides validating the efficacy of HACK in learning prototypicality, the authors also explored its use in sample complexity reduction and model robustness aspects.

**Weaknesses:**

I think the presentation of this paper needs improvements. One main issue is that the authors keep talking about how HACK works and how it can encode both visual similarity and prototypicality, without enough explanations about the reason why. It's suggested to list the intuitions upfront, so readers won't always question why HACK is designed this way and why it works at all. Specifically,
- Missing intuition everywhere about why images should be assigned to uniformly distributed particles. Only until Section 4.2, it's mentioned that this is to achieve maximum instance discrimination as in (Wu et al., 2018).
- Follow-up questions: is such uniform target the best option? Ablations on other targets will help.
- Missing another intuition: why prototypicality will merge from optimizing for maximum instance discrimination? This is never explained but super important.
- Figs 5,6,8 are supposed to show evidence that HACK indeed captures 1) visual similarity. Unfortunately I don't have the same observations from the very small image examples. Clearer examples will help. Also, image retrieval experiment is an important alternative. 2) prototypical examples (in the center of the Poincare ball) vs. atypical examples around the boundary. Again, such trend is not clear from the given small image examples.

**Questions:**

Questions around reducing sample complexity:
- Fig.9(a) shows that models trained on atypical examples performs better than on typical examples, especially when the amount of training examples used is small. This is a bit counter-intuitive and different from many other studies, where DNNs are shown to pick up regularities in typical data and then further benefit from or memorize noise/atypical data. Any comments?
- Fig.9(a) shows that with increasing amount of data (either typical or atypical) converges to similar test accuracy. Is that close to the optimal accuracy, or performance will keep improving with more data? Another (maybe more practical) way to prove sample complexity reduction is to compare to the "best" model performance and measure how much less data are used, rather than in the low-data regime where performance is far from ideal.

Questions around robustness:
- Fig.9(a) basically shows "more atypical data, better generalization accuracy", while Fig.9(b) says that using fewer atypical data improves model robustness. The observations are a bit contradicting and it seems hard to strike the balance between accuracy and robustness. Any comments?

---

### Official Review · Reviewer_qJ86 · 2023-11-01

**Soundness:** 3 good
**Presentation:** 3 good
**Contribution:** 3 good
**Rating:** 5
**Confidence:** 2

**Summary:**

The paper introduces a novel approach to map images into a feature space that not only indicates visual similarity but also encodes the prototypicality of the image based on its location in the dataset. Instead of using Euclidean space, the authors utilize hyperbolic space for unsupervised feature learning. In this space, the proximity of a point to the origin signifies its prototypicality. They present an algorithm called HACK, which assigns each image to uniformly packed particles in hyperbolic space, optimizing the dataset's organization. The method grounds the concept of prototypicality in congealing, aligning images to appear more common and similar, which aligns with human visual perception. The paper's contributions include the first unsupervised feature learning method capturing both visual similarity and prototypicality, and the demonstration that identified prototypical and atypical examples can optimize sample complexity and model robustness.

**Strengths:**

Strength:
1. Paper is well organized.
2. The use of hyperbolic space instead of Euclidean space is well-motivated.

**Weaknesses:**

Weakness:
1. CIFAR and MNIST are too toy. ImageNet experiment and fair comparison with previous unsupervised learning (especially contrastive learning) are important, but missing in this work.
2. LeNet is also too toy for a fair comparison with the latest results on unsupervised learning. A model of the ResNet level is a must.
3. Some related works on prototype learning are not cited, like “Prototypical Contrastive Learning of Unsupervised Representations”.

**Questions:**

Questions:
1. In terms of optimization, the proposed method also needs to alternatively optimize the encoder (θ) and the assignment (π), which show no advantage over previous “prototype contrastive learning work” that requires to optimize both sample features and prototype assignments ("centroids")
2. “pack the particles into a two-dimensional hyperbolic space” Is it possible to expand the embedding space to over two dimensions? I believe representing high-dimensional data into a two-dimensional space is too limited for practically useful embeddings.

---

### Official Review · Reviewer_PGEd · 2023-11-02

**Soundness:** 2 fair
**Presentation:** 2 fair
**Contribution:** 2 fair
**Rating:** 5
**Confidence:** 4

**Summary:**

In this paper, the authors propose an unsupervised feature learning algorithm, HACK, that captures visual similarity and prototypicality. Specifically, HACK first generates uniformly packed particles in the Poincare ball of hyperbolic space. Then, it optimizes data assignments to a uniformly distributed particle set by naturally exploring the properties of hyperbolic space, in which prototypical and semantic structures of data emerge finally.

**Strengths:**

In this work, the authors propose the unsupervised feature learning method from a novel perspective that aims to capture both visual similarity and prototypicality.

**Weaknesses:**

1.	The motivation of the proposed method is not clear. It lacks clarification of motivation to state that: what are the shortcomings of existing methods that do not consider prototypicality? Why does the unsupervised feature learning method need to consider prototypicality? The motivation mentioned in the first paragraph of Section 1 is too vague.
2.	In the paper, the work has limited motivation, which seems to be a combination of existing technologies with introducing existing concepts.
3.	It is also necessary to analyze the unique points of the proposed method compared to existing related methods, so as to further clarify the motivation and novelty. However, the paper lacks concrete analyses of the difference between the proposed and existing related methods.
4.	The writing of this paper needs to be improved. Some sentences include too many prepositions, which decreases readability.
5.	The layout of the article needs to be improved, for example, there is too much white space on page 8.

**Questions:**

Please see the Weaknesses.